# Full Stage Learning to Rank: A Unified Framework for Multi-Stage Systems

## ABSTRACT

The Probability Ranking Principle (PRP) has been considered as the foundational standard in the design of information retrieval (IR) systems. The principle requires an IR module's returned list of results to be ranked with respect to the underlying user interests, so as to maximize the results' utility. Nevertheless, we point out that it is inappropriate to indiscriminately apply PRP through every stage of a contemporary IR system. Such systems contain multiple stages (e.g., retrieval, pre-ranking, ranking, and re-ranking stages, as examined in this paper). The *selection bias* inherent in the model of each stage significantly influences the results that are ultimately presented to users. To address this issue, we propose an improved ranking principle for multi-stage systems, namely the Generalized Probability Ranking Principle (GPRP), to emphasize both the selection bias in each stage of the system pipeline as well as the underlying interest of users. We realize GPRP via a unified algorithmic framework named Full Stage Learning to Rank. Our core idea is to first estimate the selection bias in the subsequent stages and then learn a ranking model that best complies with the downstream modules' selection bias so as to deliver its top ranked results to the final ranked list in the system's output. We performed extensive experiment evaluations of our developed Full Stage Learning to Rank solution, using both simulations and online A/B tests in one of the leading short-video recommendation platforms. The algorithm is proved to be effective in both retrieval and ranking stages. Since deployed, the algorithm has brought consistent and significant performance gain to the platform.

## KEYWORDS

Multi-stage systems, retrieval, ranking, learning to rank

**ACM Reference Format:**
Anonymous Author(s). 2018. Full Stage Learning to Rank: A Unified Framework for Multi-Stage Systems. In *Proceedings of Make sure to enter the correct conference title from your rights confirmation emai (Conference acronym 'XX).* ACM, New York, NY, USA, 10 pages. https://doi.org/XXXXXXX.XXXXXXX

## 1 INTRODUCTION

Information retrieval (IR) systems widely influence numerous aspects of our daily lives, from information search to entertainment choices and travel decisions. The significant impact has gathered extensive research interest and gained substantial attention since its inception [31]. Central to this research area is the precise inference of users' underlying information needs, so as to present the

most relevant information to the users, a concept that proved to be optimal by the Probability Ranking Principle (PRP) [28].

Early year's IR research only concerned single-stage systems, for example using BM25 [27] or a logistic regression model [12] to rank results against a given query or user. Propelled by the emergency of new demands and development of technologies, modern IR systems nowadays are equipped with multiple stages, such as retrieval and ranking stages employed in YouTube [7], the recall, first round ranking, and second round ranking stages in Yahoo search [36], and the retrieval, pre-ranking, ranking and re-ranking stages in KuaiShou [34]. No matter it is manually crafted rules, such as the BM25 scoring function, or data-driven rankers, such as learning to rank models [23], in each component of these multi-stage IR systems, PRP is still the design principle, i.e., each stage is supposed to present the most relevant results for the next stage's further processing.

However, the optimality of PRP strongly depends on the assumption that the top-ranked result by the algorithm will eventually be presented to the end users. This is only true in a single-stage IR system, but no longer holds in multi-stage systems. For example, at the retrieval stage (typically an early stage), even if we know the underlying preferences of users on each result in the whole candidate set and return the most relevant ones to the later stage, there is a possibility that some or even all of them are filtrated eventually because the *selection bias* of subsequent stages can deviate from user's underlying interests and promote sub-optimal results along the way until reaching the end users [1].

The fundamental reason for the *selection bias* resides in the misalignment between how the ranking models in each stage are learned and how they are used in a multi-stage system. Typically, the training data for each stage is constructed by treating the results from this stage and finally preferred by the end users as positive and the rest from this stage as negative. Sub-sampling can be devised when the stage returns a large number of results (e.g., more than thousands), but the end users can only interact with a few (e.g., less than ten). However, relevant results from the intermediate stage but eventually filtered by later stages cannot be differentiated from those truly irrelevant ones, and are often mistakenly treated as negatives. This phenomenon is analogous to position bias in user feedback [9, 17], where non-clicked results do not suggest irrelevance, as they could also not be examined by a user. In a multi-stage system, the selection bias is introduced by the subsequent components in an IR pipeline, which is more implicit and dynamic. Hence, those well-known solutions for correcting position bias [6, 13, 18, 30] do not apply to this problem.

Moreover, the operation and management of multi-stage systems in industry practice further exacerbates the selection bias. Due to the system's high complexity, each stage or even a particular algorithm at a single stage is often managed by a different team. When improving an algorithm or choosing which algorithm to

---

[1] In a commercial system, there can be tens of different ranking algorithms employed in each stage of the pipeline, making the selection bias inevitable.

deploy, each team can only access and control data at its specific stage. However, these intermediate algorithmic decisions are only informed by online A/B tests on the final results.

In this paper, we propose an improved version of PRP named the Generalized Probability Ranking Principle (GPRP) for multi-stage IR systems. GPRP extends PRP by explicitly modeling the ranking preferences from both the end users and subsequent stages. GPRP degrades to PRP when there only one stage is present or it functions at the final stage of the system. At non-final stages, the expected ranking utility achieved by GPRP is always an upper bound of the expected utility achieved by PRP, which can also be proved to be a max-min approximation of GPRP. However, it is difficult to precisely fulfill GPRP in practice due to computational complexity. Under mild assumptions about a multi-stage IR system, we develop an efficient and effective algorithmic framework named Full Stage Learning to Rank (FS-LTR) to realize GPRP approximately. The core idea of FS-LTR is to define preferential treatments on the exposure data collected in each stage of the system as well as users' feedback for the final ranking and fit an LTR algorithm with the preferential labels to estimate the most effective ranking for each particular stage. FS-LTR can be seamlessly applied to any stage in the entire pipeline. Extensive experiments on both a simulated environment and online A/B testing in one of the world's largest short-video recommendation platforms validate the effectiveness of our framework. To the best of our knowledge, we are the first to study ranking principles for multi-stage IR systems holistically and to design a universally efficient algorithmic framework with theoretical foundations.

## 2  RELATED WORK

**Retrieval Stage:** Retrieval is the first stage of the recommendation system, which needs to recall users' interested items from a large candidate pool. There are various different retrieval algorithms including basic dual-encoder model [3, 7, 21], tree-based deep models [41, 42], and recent generative retrieval [32, 33]. In-batch softmax loss and Bayesian Pairwise Loss (BPR) are two frequently used losses in common retrieval algorithms, and training data mainly contain items with positive feedback and random items sampled from the whole candidate set.

**General Ranking Stage:** This part can be more carefully divided to three (or even more) stages, including pre-ranking [22, 34], ranking [4, 5, 7, 25, 39, 40], and re-ranking [1, 2, 10, 16, 24]. The difference between models in the three stages mainly lies in their model capacity and training method, either in a pointwise, pairwise, or listwise manner.

**Unbiased Learning:** The goal of unbiased learning is to eliminate the exposure bias introduced by training data collected from recommendation systems and learn unbiased ground-truth interests of users. one main idea of unbiased learning is based on the Inverse Propensity Score (IPS) estimator [29], which is used either at the retrieval stage [6] or re-ranking stage [18, 30].

## 3  PRELIMINARY OF MULTI-STAGE SYSTEMS

Without loss of generality, we consider a multi-stage system with four stages: retrieval (stage 1, from all candidates to $c_1$ items), pre-ranking (stage 2, from $c_1$ items to $c_2$ items), ranking (stage 3, from $c_2$ items to $c_3$ items), and re-ranking (stage 4, from $c_3$ items to $c_4$ items),

where $c_1, c_2, c_3, c_4$ are predefined values like 10000/500/50/6 respectively. Note the output of one stage is exactly the input candidate set of the next stage. Finally, $c_4$ items outputted by the re-ranking stage are exposed to a user and receive corresponding user feedback. Given a candidate item set $A$, a real-valued function $f(u, v)$ and a positive integer $c$, the **Topk**$(A, f(u, v), c)$ operator returns the top $c$ items in the item set $A$ in the descending order of $f(u, v)$.

Denote $u \in \mathbb{R}^d$ as the user (or query) representation, and $v \in \mathcal{I}$ as the item representation, where $\mathcal{I}$ is the whole candidate set. Once an item $v$ is exposed to a user $u$, we observe corresponding Bernoulli feedback $Y(u, v) \in \{0, 1\}$, which is sampled from the underlying interest $r(u, v) \in [0, 1]$, thus $\mathbf{E}[Y(u, v)] = r(u, v)$.

Under the above four-stage system, given a user $u$, denote the corresponding candidate set and output item set of stage $i$ as $\mathcal{I}^{i-1}(u)$ and $\mathcal{I}^i(u)$ respectively, where $i \in [4] := \{1, 2, 3, 4\}$ and $\mathcal{I}^0 \equiv \mathcal{I}$, $|\mathcal{I}^i(u)| = c_i$. For the clarity of our notations, we will use $\mathcal{I}^i$ instead of $\mathcal{I}^i(u)$ when there is no ambiguity. Given candidate set $\mathcal{I}^{i-1}$ at stage $i$, suppose item $v \in \mathcal{I}^{i-1}$ is selected into the output item set $\mathcal{I}^i$ with probability $p^i(u, v|\mathcal{I}^{i-1})$, and denote $O^i(u, v|\mathcal{I}^{i-1})$ as the corresponding observed Bernoulli random variable. Thus $\mathbf{E}[O^i(u, v|\mathcal{I}^{i-1})] = p^i(u, v|\mathcal{I}^{i-1})$. Here, we want to emphasize that $p^i(u, v|\mathcal{I}^{i-1})$ depends on $\mathcal{I}^{i-1}$, which means the probability of the same item that enters the next stage is also influenced by other candidate items in the same stage and reveals the combinatorial complexity of real-world systems.

## 4  GENERALIZED PROBABILITY RANKING PRINCIPLE

Before we introduce Generalized Probability Ranking Principle (GPRP), we recap the Probability Ranking Principle (PRP) in classical information retrieval [28, 35] first:

*Probability Ranking Principle (PRP): If an Information Retrieval system's response to each request is a ranking of items in the collections in order of decreasing probability of usefulness to the user who submitted the request, then the overall effectiveness of the system to its users will be the best that is obtainable on the basis of the data.*

The principle is intuitive and proved from the view of traditional measure of effectiveness and decision theory under certain assumptions [28]. Suppose we know the underlying interest function $r(u, v)$ in advance for every request and the system has only one stage (using the same notation as previous section that the system needs to return $c_4$ items to each request), according to PRP, then $c_4$ items returned by the system should be:

$$\underset{\{v_k \in \mathcal{I} | k \in [c_4]\}}{\mathrm{argmax}} \sum_{k=1}^{c_4} r(u, v_k) \quad (1)$$

Suppose we are in the 4-stage system described before, and we need improve a particular stage $i \in \{1, 2, 3, 4\}$ but have no control on other stages. If we still use PRP in stage $i$, we should output:

$$\mathcal{I}^{i, PRP} := \underset{\mathcal{I}^i \subset \mathcal{I}^{i-1}}{\mathrm{argmax}} \sum_{v \in \mathcal{I}^i} r(u, v) \quad (2)$$

Hence, nearly all previous studies in information retrieval systems focus on how to learn the underlying interest function $r(u, v)$ from logged exposure data, for either retrieval stage [3, 7, 19, 21, 42] or ranking stage [7, 23]. Furthermore, since logged exposure data is affected by selection bias of the system which may lead to biased

interest in learned models, some work further studies how to eliminate selection bias of the system to learn users' unbiased interests [6, 13, 18, 30, 38].

However, even though we could learn $r(u, v)$ perfectly, in a multi-stage system, the item set $\mathcal{I}^i$ ($i < 4$) cannot be exposed to users directly and it needs to be further filtrated by subsequent stages. In other words, even if we could find some items of high relevance to a user, once these items are filtered by subsequent stages, it would not bring any benefit since users can not see them. Thus, what we really care about in stage $i$ is the usefulness of final returned items after all subsequent stages, which means we should take both the preference of subsequent stages and the true interest of users into consideration at the same time. Hence we propose the following Generalized Probability Ranking Principle:

*Generalized Probability Ranking Principle (GPRP): In the stage $i$ (any $i \in \{1, 2, 3\}$) of a four-stage system, if the output of stage $i$ to each request is a set which considers both the preference of subsequent stages and usefulness to the target user, as stated in the following equation (3), then the overall effectiveness of that stage to its users will be the best that is obtainable on the basis of the data.*

$$\mathcal{I}^{i,GPRP} := \underset{\mathcal{I}^i \subset \mathcal{I}^{i-1}}{\text{argmax}} \, \mathbf{E}[\sum_{v \in \mathcal{I}^4(\mathcal{I}^i)} r(u, v)] \qquad (3)$$

where $\mathcal{I}^4(\mathcal{I}^i)$ is a random set that represents the finally exposed item set $\mathcal{I}^4$ obtained from the candidate set $\mathcal{I}^i$ after stages $i+1, \ldots, 4$, and the expectation is over the randomness of subsequent stages.

According to the definition of equation (3), it's easy to see the expected utility of PRP is always a lower bound of the expected utility of GPRP when deployed in online systems:

$$\mathbf{E}[\sum_{v \in \mathcal{I}^4(\mathcal{I}^{i,PRP})} r(u, v)] \leqslant \mathbf{E}[\sum_{v \in \mathcal{I}^4(\mathcal{I}^{i,GPRP})} r(u, v)] \qquad (4)$$

To certain extend, the core philosophy behind GPRP is on the opposite to unbiased learning discussed in Section 2. Both of them consider the selection bias of the system. However, the goal of unbiased learning is to eliminate selection bias and learn the underlying interest function of users, i.e. $r(u, v)$, which is then used to make decisions. While in GPRP of a multi-stage system, though underlying interest function $r(u, v)$ is important, the selection bias of subsequent stages is also important in determining returned items, and we should consider both of them when making decisions, which will lead to better final performance according to inequality (4).

However, it is hard to formulate random set $\mathcal{I}^4(\mathcal{I}^i)$ given the fact that subsequent stages after stage $i$ may contain both black-box algorithmic models and manual filtering strategies in practical systems, let alone solving equation (3). Hence, we should consider a simplified approximation of equation (3) which allows efficient algorithms. We find that PRP can be viewed as a max-min or conservative approximation of equation (3) as stated in the following proposition, which suggests we can completely ignore the complex process of subsequent stages and optimize its minimal utility in the worst case scenario:

PROPOSITION 1. $\mathcal{I}^{i,PRP}$ *defined in equation (2) is also the solution to the following optimization problem:*

$$\underset{\mathcal{I}^i \subset \mathcal{I}^{i-1}}{\text{argmax}} \, \underset{\{v_k \in \mathcal{I}^i | k \in [c_4]\}}{\min} \sum_{k=1}^{c_4} r(u, v_k) \qquad (5)$$

Because of space limitations, all the proof details are presented in the appendix.

We are interested in studying the gap between the final utility from PRP and GPRP, which is defined as

$$\mathbf{E}[\sum_{v \in \mathcal{I}^4(\mathcal{I}^{i,GPRP})} r(u, v)] - \mathbf{E}[\sum_{v \in \mathcal{I}^4(\mathcal{I}^{i,PRP})} r(u, v)] \qquad (6)$$

Apparently, when the system has only one stage or in the last stage of a multi-stage system, there are no subsequent stages, and the returned item set can be exposed to uses, hence GPRP degrades to PRP and the performance gap (6) is 0. But we find the gap can be large in a non-final stage of a multi-stage system in the worst case without any assumption, as stated in the following proposition:

PROPOSITION 2. *The performance gap (6) between PRP and GPRP can be $c_4$ in the worst case.*

Note the upper bound of equation (6) is exactly $c_4$, which means PRP is not a good approximation of GPRP in the worst case of a multi-stage system and thus we have to find some other proxy for the objective function in equation (3).

Since $\mathcal{I}^4(\mathcal{I}^i)$ is a random set depending on the input candidate set, and the probability of each item $p^i(u, v | \mathcal{I}^{i-1})$ being selected into the next stage $i + 1$ is also influenced by other candidates, which causes the computational complexity of solving equation (3) exactly. We then make the following assumption to allow efficient approximate solution to equation (3):

ASSUMPTION 1. *Assume the selection probability $p^i(u, v | \mathcal{I}^{i-1})$ depends only on the user and item itself, i.e. $p^i(u, v | \mathcal{I}^{i-1}) = p^i(u, v)$.*

This is a rather strong assumption in practical systems at first glance, since general ranking stages need to compare all input candidates together, hence we cannot ignore the influence of other candidate items in each request. We will discuss the validity of this assumption in detail in Section 6.

Under Assumption 1, we can rewrite the objective function (3) of GPRP into the following:

$$\bar{\mathcal{I}}^{i,GPRP} := \underset{\mathcal{I}^i \subset \mathcal{I}^{i-1}}{\text{argmax}} \sum_{v \in \mathcal{I}^i} p^{i+1,4}(u, v) r(u, v) \qquad (7)$$

where $p^{i,j}(u, v) := p^i(u, v) p^{i+1}(u, v) \cdots p^j(u, v)$. Here we use $\bar{\mathcal{I}}^{i,GPRP}$ to distinguish $\mathcal{I}^{i,GPRP}$ in equation (3). When Assumption 1 holds, they are the same, otherwise they are different.

Compared with the original objective function (3), the new objective function looks more like the objective function (2) of PRP, which gets rid of the annoying combinatorial form of subsequent stages and allows efficient learning like many previous algorithms under PRP. However, different from the objective function (2) of PRP which only considers the interest of users, the new objective function (7) still considers both the selection bias of subsequent stages (i.e. $p^{i+1,4}(u, v)$ term) and the interest of users (i.e. $r(u, v)$ term), which inherits the merit of GPRP.

Without Assumption (1), though it is hard to compare $\bar{\mathcal{I}}^{i,GPRP}$ in (7) and $\mathcal{I}^{i,PRP}$ in (2) in general, we give two observations in some special cases of the system, which may help us understand the relation between them.

OBSERVATION 1. *If the selection bias of subsequent stages $p^{i+1,4}(u, v)$ is independent of items (or say without selection bias), then $\bar{\mathcal{I}}^{i,GPRP} = \mathcal{I}^{i,PRP}$.*

The observation is obvious. For example, when subsequent stages' output candidate items to the next stage are completely random, then the optimal candidate set should be chosen according to the underlying interest of users. However, the completely random strategy is very dangerous in an online environment which can hurt users' experience on the platform, and the strategy in each stage is often some learned models from logged data.

OBSERVATION 2. *If the selection bias of subsequent stages $p^{i+1,4}(u,v)$ is monotonically increasing with respect to the underlying interest $r(u,v)$ , then $\bar{I}^{i,GPRP} = I^{i,PRP}$.*

Above observation is also straightforward, since $p^{i+1,4}(u,v)$ is monotonically increasing with $r(u,v)$, which implies the ranking of items in terms of $p^{i+1,4}(u,v)r(u,v)$ is the same as the ranking by $r(u,v)$, thus $\bar{I}^{i,GPRP}$ equals $I^{i,PRP}$. However, the assumption in observation 2 is too strict, as it requires the preference of each stage to exactly coincide with the interest of users in every request, which is impractical considering the existence of learning error of any algorithm in any stage.

Now, given the difference between objective functions (7) and (2), the key question is then how to solve equation (7) efficiently. A straightforward method is to learn $p^{i+1,4}(u,v)$ and $r(u,v)$ separately, where each term in $p^{i+1,4}(u,v)$ can be learned by any supervised algorithm on corresponding data and $r(u,v)$ term can be learned by any existing algorithm especially unbiased learning algorithms [6, 13, 18, 30, 38]. Having learned $p^{i+1,4}(u,v)$ and $r(u,v)$, we can sort items in $I^{i-1}$ according to $p^{i+1,4}(u,v)r(u,v)$ and output the top $c_i$ items as $\bar{I}^{i,GPRP}$. However, this method is also inefficient and may be ineffective, as we need to learn several models which causes additional computational burden and accumulated learning error. Besides, in the retrieval stage, usually we use Approximate Nearest Neighbor (ANN) for fast inference, which does not support the operation in equation (7). To address the above issue, we propose a unified algorithmic framework named Full Stage Learning to Rank.

## 5 FULL STAGE LEARNING TO RANK

According to objective function (7), we need to sort items according to $p^{i+1,4}(u,v)r(u,v)$. Instead of learning each term independently, we directly learn the product $w^{i+1,4}(u,v) := p^{i+1,4}(u,v)r(u,v)$ to avoid issues mentioned in previous section. Apparently $O^{i+1,4}(u,v)Y(u,v)$ is a realization which can be observed as $w^{i+1,4}(u,v)$ in practical systems. As $\mathbf{E}[O^{i+1,4}(u,v)Y(u,v)] = w^{i+1,4}(u,v)$, we obtain our first efficient unbiased algorithm for learning objective function (7) of GPRP under Assumption 1: we collect data after stages $i-1$ with label 0, except the exposed and clicked data with label 1, then learning a supervised model with such labeled data which is then used online at stage $i$.

Though this algorithm is simple and easy to implement, it treats all non-exposed items equally, which may omit the abundant information of the data. Intuitively, items which enter stage $i+1$ may be better than items filtrated by stage $i$, thus we hope to make full use of the information of collected data to learn the selection bias and user's interest better.

Before introducing our new algorithmic framework, we make another assumption about the multi-stage system that is essential for later analysis:

ASSUMPTION 2. *For two observed items $v_i, v_{i+1}$ in two subsequent stages $i$ and $i+1$ (i.e. $v_i \in I^i, v_i \notin I^{i+1}, v_{i+1} \in I^{i+1}$), where $i \in [3]$,*

*the ratio between user's interests of these two items is bounded in a constant interval, which is less than the ratio of selecting probability between them, that is*

$$\frac{1}{a} \leqslant \frac{r(u,v_{i+1})}{r(u,v_i)} \leqslant a \leqslant \frac{p^{i+1:4}(u,v_{i+1})}{p^{i+1:4}(u,v_i)} \tag{8}$$

*where $a \geqslant 1$ is a constant.*

The intuition behind Assumption 2 is that each stage of the system can be seen as a cluster of items depending on the degree of user's interest in some sense. Apparently, for an information retrieval system with good performance, user's interest for items in higher stages may be stronger than interest for items in lower stages with high probability. What's more, the interest of users with respect to two items in subsequent stages are close to each other (i.e. $\frac{r(u,v_{i+1})}{r(u,v_i)} \in [\frac{1}{a}, a]$). Note this condition does not require the selection bias of the system is exactly the same with user's underlying interest, and allows the learning error of system to some degree, which is much weaker than the monotone assumption of multi-stage systems used in Observation 2 and in line with practical situation. See more discussion about this assumption in next section, where we collect real-world online data from one of largest short-video platforms to verify it in an approximate way and the result implies Assumption 2 is relatively mild in practical multi-stage recommendation systems.

For $t$-th request of the system, we collect a series of data at all stages $\{(u^t, v_k^t, S_k^t, \bar{Y}_k^t)|k \in [M]\}$, where $M$ is the number of collected data in each request and $S_k^t \in [4]$ is an observed random variable representing the stage of item $v_k^t$. $\bar{Y}_k^t \in \{0, 1, NA\}$ here represents the feedback of $(u, v_k)$ pair, where $\bar{Y}_k = Y_k$ when $S_k = 4$, otherwise $\bar{Y}_k = NA$, since we can only observe the true feedback $Y_k(u, v_k)$ when the item is exposed to the user.

With these data collected from full stages of different requests, the main technique of our method is to relabel each user-item pair $(u, v, S, \bar{Y})$ in collected data set by the following rule:

$$L(u,v) = \begin{cases} z_0 & \text{if } S(u,v) = 0 \\ z_i & \text{if } S(u,v) = i, \text{for } i < 4 \\ z_4 & \text{if } S(u,v) = 4, \text{and } Y(u,v) = 0 \\ z_5 & \text{if } S(u,v) = 4, \text{and } Y(u,v) = 1 \end{cases} \tag{9}$$

where $z_0 \leq z_1 \leq \cdots \leq z_5$ are six non-negative numbers.

The core idea behind this relabeling technique is intuitive, which distinguish the difference among non-exposed items and makes full use of them. Items which are returned to users and receive positive feedback apparently are the most important ones, since they have passes the examination of selection bias of subsequent stages and enjoys users interest. For those items that have been retrieved but not shown to users, entering into next stage of the system is more difficult compared with being liked by users. Therefore, if an item enters into a higher stage of the system, it is more important compared with items in lower stages. In fact, under Assumption 1 and 2, one can prove above relabeling technique coincides with GPRP as stated in the following theorem, which give us a theoretical support of our simple relabeling technique.

THEOREM 1. *Suppose Assumption 1 and 2 hold in multi-stage systems, for any collected request data $\{(u, v_k, S_k, \bar{Y}_k)|k \in [M]\}$, after relabeling via equation (9), we obtain new labels $L(u, v_k)$ for each data respectively, the a ranking of items in decreasing order of $L(u, v_k)$ implies the ranking by term $w^{i+1,4}(u, v_k)$ for items in*

---

**Algorithm 1:** Full Stage Learning to Rank

---

1 **Training:**

**Input:** Initialized model parameter $\theta$, manually defined labels $\{z_i | i \in [0, 5]\}$ and original data-set $\{\{(u^t, v_k^t, S_k^t, \bar{Y}_k^t) | k \in [M], S_k^t \geq j\} | t \in [N]\}$, where $j \in [0, i+1]$ is a hyper-parameter.

**Output:** Model $\hat{l}(u, v | \theta)$

2     **While $\theta$ not converged do**

3         sample a batch of requests from $[N]$

4         relabel each user-item pair by the equation (9)

5         update model parameters by chosen LTR algorithm

6     **return $\theta$**

7 **Serving in stage $i$:**

**Input:** Candidate set $\mathcal{I}^{i-1}$ in a request of user $u$, learned model parameters $\theta$

**Output:** $\mathcal{I}^i$

8     $\mathcal{I}^i := \textbf{Topk}(\mathcal{I}^{i-1}, \hat{l}(u, v | \theta), c_i)$

9     **return $\mathcal{I}^i$**

---

*different stages. As a special case, when $z_0 = z_1 = \cdots = z_4 = 0$ and $z_5 = 1$, the conclusion still holds even without Assumption 2.*

According to Theorem 1, now we can use the new label $L(u, v)$ as an approximate substitution of $w^{i+1,4}(u, v)$, and this new relabeling also allows efficient learning. Once we can estimate $L(u, v)$ or has the ability of ranking items which is the same as ranking by $L(u, v)$, then it nearly matches GPRP, since $\mathcal{I}^{i,GPRP}$ is obtained by ranking $w^{i+1,4}(u, v)$ in decreasing order according to equation (7) under Assumption 1. Thus, we can use any supervised or general ranking algorithm to reach an efficiently learned model that coincides with GPRP. In detail, having obtained the collected relabeled data $\{\{(u^t, v_k^t, L_k^t) | k \in [M]\} | t \in [N]\}$, where $N$ is the total number of collected requests, we can use any supervised learning or Learning to Rank (LTR) algorithm to learn a model with the goal of correctly ranking items $\{(u^t, v_k^t, L_k^t) | k \in [M]\}$ in each request, for example Lambda Rank [1, 2].

At stage $i$, since we only care about the selection bias of stages after $i$ and users' underlying interest, the collected data at stage $i$ and before stage $i$ seems useless. However, from the view of consistency between training and serving, it will be better to use them duration training. For example when serving at retrieval stage, we use the learned model to predict scores for all items. Therefore, it will help a lot to add some random samples from the candidate pool during training. The same reasoning applies at other stages. What's more, we can also add the collected data at any other previous stage duration training, which can be regarded as an auxiliary training task and may help learn the model better. Note using data in previous stages doesn't influence the near consistency between our algorithm and GPRP.

The final algorithmic framework including training and inference is given in Algorithm 1. Since we need collect data from full stages of a system and use LTR algorithm as our backbone, we name our algorithmic framework as Full Stage Learning to Rank, FS-LTR in short. When $S_k^t = 0$ in Algorithm 1, it means the item $v_k^t$ is randomly sampled from $\mathcal{I}$.

As discussed above, by choosing different $j$ and appropriate models, Algorithm 1 can be used either in retrieval, pre-ranking, ranking or re-ranking. For example, in retrieval stage, we can choose $j = 0$ which includes random negative items from the whole candidate set and dual-encoder models which represents users and items as vectors and supports ANN for fast inference. Similarly in pre-ranking/ranking stage, we can choose suitable $j$ and dual-encoder model or complex dnn models. In re-ranking stage, Algorithm 1 degrades to backbone LTR algorithm if only using exposed data, and becomes a new LTR algorithm with auxiliary tasks which uses data in previous stages.

## 6 DISCUSSION ABOUT ASSUMPTIONS

As mentioned in Section 4, Assumption 1 is too strong to be true in real complex multi-stage systems, but it is acceptable if this assumption could be satisfied in some degrees, and we explain its reasonableness from four viewpoints:

1. The simplification from a combinatorial system into point-wise system is quite common method in similar situation, like in re-ranking [14].

2. Most of previous work about unbiased learning [6] are based on estimating point-wise exposure probability to selection exposure bias of the system and then learn the underlying interests, which also rely on Assumption 1 implicitly.

3. Though the probability $p^i(u, v | \mathcal{I}^{i-1})$ varies with input candidate set $\mathcal{I}^{i-1}$, we may consider the general performance of each user-item pair, i.e. $p^i(u, v) = \mathbf{E}[p^i(u, v | \mathcal{I}^{i-1})]$ as an approximation to $p^i(u, v | \mathcal{I}^{i-1})$, where the expectation is over some distribution of $\mathcal{I}^{i-1}$ (for example the distribution of $\mathcal{I}^{i-1}$ of current system).

4. We only use Assumption 1 to induce efficient and practical algorithm. In experimental section, we find our algorithm still works when this assumption does not hold either in simulated experiments or online A/B testing.

Assumption 2 is also critical for FS-LTR. However, it is impossible to collect real $r(u, v)$ and $p^i(u, v)$ in real systems, as we cannot present the same item twice for a user in short-video platform which implies we can only observe one realization $Y(u, v)$ of underlying Bernoulli distribution with probability $r(u, v)$. What's more, items at non-final stages of the system are not exposed to users, so we cannot even observe their realizations of users' underlying interest. Besides, practical system is very complex, there does not exist any true $p^i(u, v)$ in real situation.

One possible approach to solving above difficulty is to learn approximate models about $r(u, v)$ and $\{p^i(u, v) | i \in [4]\}$ respectively. However, this approach highly depends on the learning performance which is also hard to have some guarantee, because of the discrepancy between learning space and inference space. Therefore, we adopt an approximate verification approach.

In detail, we collect items at different stages in previous request (there are 5 stages in our system), and then force these items to be exposed to corresponding users directly, which don't need to enter the multi-stage system to avoid its selection bias. Now we can receive ground-truth feedback of items at different stages. Table 1 shows the average performance of CTR (Click Through Rate) at different stages of such collected data. We can see average posterior CTR is very close between consecutive stages, and the ratio

## Table 1: Posterior CTR of Items at Different Stages.

|  | Stage 1 | Stage 2 | Stage 3 | Stage 4 | Stage 5 |
|---|---|---|---|---|---|
| *Posterior CTR* | 0.29 | 0.33 | 0.34 | 0.49 | 0.56 |

between them is also in a small interval $[\frac{1}{2}, 2]$. Thus we know users' interest for all items which have entered the system are close at least on average, which implies the inequality $\frac{1}{a} \leqslant \frac{r(u, v_{i+1})}{r(u, v_i)} \leqslant a$ of equation (8) may be true in real-world and $a$ may be 2 in our case. Besides, the inequality about term $\frac{p^{i+1:4}(u, v_{i+1})}{p^{i+1:4}(u, v_i)}$ in equation (8) means an item that could be selected to the next stage is relatively more difficult than user whether likes it compared with other candidate items in subsequent stages. In practical system where $c_1/c_2/c_3/c_4/c_5 = 6000/3000/500/120/10$, the selecting probability from stage 2 to 3 is roughly $\frac{500}{3000} = \frac{1}{6}$ on average, which means the ratio of selecting probabilities between positive (i.e. select by stage 2 into stage 3) and negative item (i.e. filtrated by stage 2) in this stage is roughly 6 on average, hence the ratio of exposure probability between these two items after subsequent stages could be even larger than 6, which may be greater than $\frac{r(u, v_{i+1})}{r(u, v_i)}$ with high probability. Therefore, Assumption 2 may be mild enough to be satisfied in a real environment, at least in a short-video platform.

## 7 PRACTICAL IMPLEMENTATION

In this section, we present our implementation details of FS-LTR in real-world online multi-stage recommendation systems, including data collection, training, and serving.

**Data Collection:** Collecting data in full stages of a recommendation system online is a very challenging engineering task, since the number of these data far exceeds the magnitude of the exposure data in each request. For example, it could be 10000 versus 6 in real system, hence it is impractical to collect all of them considering the cost of storage and communication bandwidth. In practice, for each reqeust, we randomly sample 40 negative items in retrieval and pre-ranking stage respectively, and record all items (around 400) in ranking stage with corresponding labels representing the final stage of items.

**Training:** We use lambda rank to train our model in retrieval stage with $z_i := i$. To reduce training cost, we only use a subset of collected data in each request. The higher stage the data belongs to, the more we use it, since data in higher stages is more important and hard to be learned. We also substitute the cross-entropy loss in lambda rank with margin loss, as well as trying different labels (for example, 0, 1, 2, 3, 4, 6) to enhance the learning of hard samples. Though having some improvements in offline evaluation, it doesn't lead to online gain. What's more, to speed up training, one could use pairwise LTR instead of lambda rank, which avoids expensive sort operation during training but with only mild damage in online performance. Finally, it is also possible to use softmax loss with soft labels $l(u, v)$ to further speed up training.

**Serving:** At retrieval stage, to enhance the diversity of returned results in each request, we add some noise to the top representation vector of users, which also fulfills some mild exploration. The same trick could be used in other stages too.

## 8 EXPERIMENTS

In this section, we conduct both offline experiments and online A/B testing to answer the following research questions:

**RQ1** How can we simulate a multi-stage system with an offline dataset for evaluating our proposed method? (Sec 8.1)

**RQ2** What is the effectiveness of FS-LTR on the offline simulated multi-stage recommendation system? (Sec 8.2)

**RQ3** How does each component or hyperparameter in Full Stage Learning to Rank affect the performance? (Sec 8.3.2)

**RQ4** How does FS-LTR perform in online environment? (Sec 8.4)

### 8.1 Offline Simulation of Multi-Stage Pipeline

In this section, we introduce our offline simulation of the multi-stage recommendation systems from three perspectives: dataset, multi-stage simulation, and training data.

*8.1.1 Dataset.* Different stages of a multi-stage recommendation have an exponential magnitude difference in the number of candidates, thereby the first stage needs an enormous amount of candidates. However, most of the datasets in recommendation systems are highly sparse, which leads to a limited number of available reasonable candidates. Even if we can train a model to capture the latent preference of the user, without the ground truth of user-item interaction, we still cannot determine whether the user prefers the item. The original sparse ground truth may be more sparse after passing the multi-stage pipeline.

## Table 3: Statistics of KuaiRec dataset.

|  | #User | #Item | #Interaction | Density |
|---|---|---|---|---|
| *small matrix* | 1,411 | 3,327 | 4,676,570 | 99.6% |
| *big matrix* | 7,176 | 10,728 | 12,530,806 | 16.3% |

To build a convincing simulation on the multi-stage recommendation pipeline, we utilize the fully-observed dataset **KuaiRec**[11] as our base dataset. KuaiRec is a real-world dataset collected from the recommendation logs of a video-sharing platform. "Fully Observed" means that there are almost no missing values in the user-item interaction matrix, allowing an enormous amount of available candidates with ground truth for simulating the multi-stage recommendation pipeline. There are two user-item interaction matrices in KuaiRec, named *small matrix* and *big matrix*. The statistics of the two matrices are listed in Table 3. Both of these two matrix are more dense than most of the recommendation datasets. Moreover, rich side features are provided for each user and item in the KuaiRec dataset, enabling training a good ranking model for simulating the multi-stage pipeline. All of the user and item in *small matrix* also occur in the *big matrix*, but interactions in *small matrix* and *big matrix* are excluded from each other.

*8.1.2 Multi-Stage Simulation.* We consider building a multi-stage pipeline containing three parts: retrieval, prerank, and rank. Each item must pass all these three stages to be exposed to the user. In each stage, a learned model will score each candidate item, and items with relatively high scores can be passed to the next stage. We assume that

• **Retrieval candidate pool** contains all of the items in *big matrix*( ~11000).

**Table 2: Performance comparison on our simulated multi-stage pipeline. Bold numbers represent the best results. All the numbers in the table are percentage numbers with '%' omitted. All experiments are repeated 5 times to calculate the mean and standard deviation. We conduct an unpaired t-test of Full Stage Learning to Rank and the best baseline and the improvement of Full Stage Learning to Rank is significant with $p \leq 0.05$ for all values with '*'.**

| Stage | Method | HR@20 | NDCG@20 | R@20 | P@20 | HR@50 | NDCG@50 | R@50 | P@50 |
|-------|--------|-------|---------|------|------|-------|---------|------|------|
| Retr | BPR | 15.77(±0.97) | 5.50(±0.53) | 0.55(±0.03) | 0.88(±0.05) | 35.59(±1.85) | 9.46(±0.62) | 1.41(±0.07) | 0.91(±0.04) |
| | SSM | 17.32(±1.34) | 6.02(±0.50) | 0.59(±0.05) | 0.96(±0.08) | 37.88(±0.86) | 10.16(±0.42) | 1.53(±0.05) | 0.99(±0.03) |
| | FS-RN | 18.14(±0.49) | 6.41(±0.25) | 0.63(±0.03) | 1.03(±0.03) | 38.60(±0.73) | 10.52(±0.19) | 1.61(±0.05) | 1.04(±0.03) |
| | FS-LR | 18.83(±0.58) | 6.69(±0.12)* | 0.66(±0.02) | 1.06(±0.03) | 40.75(±0.33)* | 11.07(±0.08)* | 1.72(±0.02)* | 1.10(±0.01)* |
| PR | BCE | 14.40(±1.10) | 4.93 (±0.35) | 0.48(±0.04) | 0.76(±0.06) | 32.57(±1.68) | 8.54(±0.50) | 1.24 (±0.09) | 0.79(±0.06) |
| | FS-LR | 19.56(±1.18)* | 6.74(±0.48)* | 0.70(±0.05)* | 1.13(±0.07)* | 40.16(±0.72)* | 10.95(±0.39)* | 1.79(±0.04)* | 1.15(±0.02)* |

• **Prerank candidate pool** of each user contains all of the items exposed to the user in both *big matrix* and *small matrix*(~3500).

Since we hope the learned models have a relatively strong ability to predict users' interests, we do not mind using more data to train the prerank / rank model. We use the *big matrix* with user features and item features to train the prerank / rank model. We select the two-tower DNN for the prerank model and the single-tower DNN for the rank model. Details can be viewed in Appendix A.4. All of the prerank candidates can be scored with the prerank model and the rank model, enabling us to simulate the multi-stage pipeline.

For the convenience of later training and evaluation, we model the whole multi-stage pipeline as a static **request**. In a static request, the following steps are executed in sequence:

**Step 1** For each user, 1,000 items are randomly selected from the prerank candidate pools as the **simulated prerank candidates**.

**Step 2** For each user, the top 200 items with highest prerank scores in simulated prerank candidates are selected as the **simulated rank candidates**, while the bottom 200 items with lowest scores in simulated prerank candidates are regarded as the **simulated prerank negative samples**.

**Step 3** For each user, the top 50 items with highest rank scores in simulated rank candidates are selected as the **simulated exposed candidates**, while the bottom 50 items with lowest scores in simulated rank candidates are regarded as the **simulated rank negative samples**.

**Step 4** For each user, items with positive labels are regarded as the **simulated exposed positive samples**, while items with negative labels are regarded as the **simulated exposed negative samples**.

Multiple request procedures can be repeatedly simulated to generate different multi-stage recommendation training samples. For evaluation, requests for validation or tests are separated in advance in order to avoid label leakage.

*8.1.3 Data Preparation for Training and Evaluation.* Most of the recommendation models are training on the exposed samples, e.g., the retrieval models are trained on the exposed positive samples with randomly sampled negative samples, the rank models are trained on the exposed samples. In our simulated multi-stage recommendation pipeline, models should be trained on the simulated exposed samples. From this perspective, the prerank / rank model in the simulated multi-stage pipeline is not aligned with the real settings. However, it is acceptable since we do not mind the simulated rank

model is more powerful in the simulated multi-stage pipeline. The simulated exposed samples are partially randomly generated from the simulated multi-stage pipeline, which may cause difficulty in splitting the dataset for training, validation, and testing. In order to avoid label leakage, we follow the following steps to generate simulated training data:

**Step 1** Generate a static request based on the simulated prerank / rank model and the original prerank candidate pool. The simulated exposed positive samples are reserved as positive samples in the test set, and removed from the original prerank candidate pool.

**Step 2** Generate a static request based on the simulated prerank / rank model and the prerank candidate pool without the positive samples in the test set. The newly simulated exposed positive samples are reserved as positive samples in the validation set, and removed from the prerank candidate pool.

**Step 3** Generate multiple static requests based on the simulated prerank / rank model and the prerank candidate pool without the positive samples in the validation and test set. Samples of each request can be utilized as training samples.

Complete separation of the train, validation, and test set is accomplished by the steps above. In our request-based training setting, the training request is set to a random request. Exposed samples and multi-stage samples are randomly sampled from the randomly selected request to build multiple pair-wise or single list-wise optimization objectives.

## 8.2 Performance Comparison

*8.2.1 Implementation Details.* We utilize the **MF**[20] as the model structure for all methods. We implement all of the offline experiments with Pytorch 1.13 in Python 3.8. We set the same value for basic hyperparameters. The size of the embedding is set to 64, the batch size is set to 128. L2 normalization is applied to the embedding during training and inference. We use the Adam optimizer with a learning rate of 0.001 for training for all methods.

*8.2.2 Baseline Methods.* We select multiple widely used optimization objectives as baseline methods for our experiments:

• **BPR**[26] (Beyesian Personalized Ranking). An objective models the posterior preference probability of a user-item pair based on a single sampling negative user-item pair. The default number of negative samples is 5.

**Table 4: Online A/B testing results.**

| App Usage Time Per User | Real-show | Click | Like | Follow | Forward | Comment | Watch Time |
|---|---|---|---|---|---|---|---|
| +0.12% | +0.50% | +0.69% | +0.40% | 0.74% | +0.94% | 1.08% | +0.18% |

- **SSM**[8, 15] (Sampled-SoftMax). An objective models the preference probability of user-item pairs based on multiple sampling negative negative user-item pairs. The default number of negative samples is 5.
- **BCE** (Binary Cross Entropy). An objective directly optimizes the preference probability of the user-item pair.

BPR and SSM are common baseline methods for retrieval, and negative user-item pairs for them are sampled from the global candidate pool. BCE is adopted as the baseline method for our offline experiment on prerank stage.

*8.2.3 Evaluation Metrics.* We adopt four widely used metrics for evaluation of our methods, i.e., **R** (Recall), **P** (Precision), **HR** (HitRate), and **NDCG** (Normalized Discounted Cumulative Gain). The metrics are computed on the top 20/50 matched items. As mentioned above, the different stage has a different candidate pool for evaluation. The prerank stage has a relatively smaller candidate pool than retrieval. However, metrics only have a very limited difference between the retrieval and prerank stage for our imperfect simulation on a multi-stage recommendation system.

**Table 5: Study on the number of different stage examples. The 'x,y,z' in the Negative Sampling column denotes the number of rank negative samples, prerank negative samples, and global negative samples in one training instance, respectively. Exposed negative sample size is set to 1 for all methods.**

| Negative Sampling | HR@50 | NDCG@50 | R@50 | P@50 |
|---|---|---|---|---|
| 1,1,1 | 39.12 | 10.45 | 1.68 | 1.07 |
| 1,2,1 | 40.71 | 10.73 | **1.73** | **1.10** |
| 2,1,1 | 39.67 | 10.71 | 1.68 | 1.07 |
| 1,1,2 | **40.75** | **11.07** | 1.72 | 1.10 |
| 1,1,3 | 39.24 | 10.73 | 1.63 | 1.05 |
| 1,2,3 | 39.71 | 11.00 | 1.69 | 1.09 |

*8.2.4 Overall Performance.* Performance comparison is listed in Table 2. For baseline methods and our FS-LTR methods in retrieval, the negative sample size for each training instance is set to 5 to eliminate the effect of the negative sample size. Two variants of our FS-LTR, FS-RN(RankNet) and FS-LR(LambdaRank) outperform significantly to the baseline methods in the retrieval stage. Our FS-LR method also shows an advantage over the commonly used BCE in the prerank stage.

## 8.3 Ablation Study

Now we conduct experiments to further understand the effect of multi-stage negative samples and negative sample size in FS-LTR.

*8.3.1 Effectiveness of multi-stage samples.* The complete FS-LTR in retrieval requires user-item pairs from full stages in recommendation systems. We remove user-item pairs from different stages to verify the effectiveness of each stage sample. Results are shown in Table 6. The re-labeled label is also adapted to the number of stages

left in the training samples. Removing the stage-negative samples causes the most degradation in model performance, showing that negative samples in a multi-stage recommendation system play a key role in FS-LTR.

*8.3.2 The number of samples from different stages.* We show the performance of FS-LR on the number of samples from different stages in Table 5. Results show that more negative sample sizes may have a positive effect on model performance. The ratio of different stage negative sample sizes is also critical for model performance.

**Table 6: Ablation study on different stage examples.**

| | HR@50 | NDCG@50 | R@50 | P@50 |
|---|---|---|---|---|
| w/o *exposed neg* | 37.29 | 10.03 | 1.52 | 0.98 |
| w/o *stage neg* | 35.52 | 9.72 | 1.44 | 0.93 |
| w/o *rank neg* | 39.40 | 10.74 | 1.63 | 1.04 |
| w/o *prerank neg* | 38.31 | 10.16 | 1.56 | 1.01 |
| FS-LR | 40.75 | 11.07 | 1.72 | 1.10 |

## 8.4 Online A/B Testing

We used our Algorithm 1 at the retrieval stage on one of the largest short-video platforms with implementation details described in Section 7. The A/B test lasted for six days and had influenced over 20 million users in the experiment group, reaching a significant improvement compared with the base group which already had some strong baselines like TDM [42], Multi-Interest Retrieval [37], Comi-Rec Retrieval [3] etc. As shown in Table 4, our approach achieves significant gains in many engagement metrics. What's more, our retrieval algorithm has the highest reveal ratio (16%, where the second highest reveal ratio is 6%) of any other retrieval algorithm online, which coincides with the GPRP.

## 9 CONCLUSIONS AND FUTURE WORK

We believe our work opens a new direction of research in multi-stage IR systems, which needs to take both the selection bias in multiple stages of the system and users' underlying interest into consideration. We proved the effectiveness of this solution framework in both offline experiments and online A/B test. There are several important future directions which worth indepth exploration. First, we have taken initial efforts aim to decipher the behavior of multi-stage systems, and a more comprehensive understanding will help us design more efficient and effective ranking algorithms. Second, our focus has been primarily on the alignment between our learning objective and GPRP, thus calling for the need to examine the generalization performance throughout the entire learning processes. Third, it is worthwhile to further customize the general framework for specific stages of the system for better performance. Lastly, exploring optimal solutions to handle multiple or all system stages, rather than just one, remains imperative.

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

# A APPENDIX

## A.1 Proof of Proposition 1

PROOF. Suppose $\mathcal{I}^{i-1} = \{v_1, v_2, \ldots, v_{c_{i-1}}\}$ and $r(u, v_1) \geq r(u, v_2) \geq \cdots \geq r(u, v_{c_{i-1}})$ without loss of generality. According to equation (2), it is easy to see $\mathcal{I}^{i,PRP} = \textbf{Topk}(\mathcal{I}^{i-1}, r(u, v), c_i) = \{v_i | i \in [c_i]\}$.

Denote a general $\mathcal{I}^i$ as $\{v_{j_1}, v_{j_2}, \ldots, v_{j_{c_i}}\}$, where $j_1 < j_2 < \cdots < j_{c_i}$ and $\{j_k | k \in [c_i]\} \subset [c_{i-1}]$. Then for equation (5), it is easy to see:

$$\underset{\mathcal{I}^i \subset \mathcal{I}^{i-1}}{\text{argmax}} \min_{\{v_k \in \mathcal{I}^i | k \in [c_4]\}} \sum_{k=1}^{c_4} r(u, v_k) \quad (10)$$

$$= \underset{\mathcal{I}^i \subset \mathcal{I}^{i-1}}{\text{argmax}} \sum_{k=1}^{c_4} r(u, v_{j_{c_i+1-k}}) \quad (11)$$

$$= \{v_i | i \in [c_i]\} \quad (12)$$

$$= \mathcal{I}^{i,PRP} \quad (13)$$

□

## A.2 Proof of Proposition 2

PROOF. We construct a concrete example to prove this proposition. Without loss of generality, suppose we are at stage 3, and the final stage is 4. Let $c_2 = 4, c_3 = 3, c_4 = 1$. Suppose candidate items are $(v_1, p_1^4 = 0.1, r_1 = 1.0), (v_2, p_2^4 = 0.2, r_2 = 1.0), (v_3, p_3^4 = 0.5, r_3 = \epsilon), (v_4, p_4^4 = 0.0, r_4 = 0.0)$, where $\epsilon$ is a very small constant. Suppose the strategy at stage $i$ is to output the item with highest $p^4$. Then according to PRP, we should choose $\{v_1, v_2, v_3\}$ as the output of stage 3 and it is easy to see corresponding utility is $\epsilon$. While according to GPRP, we should choose $\{v_1, v_2, v_4\}$ as the output with utility 1. Thus the gap between them is $1 - \epsilon$. When $\epsilon$ approaches 0, the gap approaches $c_4 = 1$. □

## A.3 Proof of Theorem 1

PROOF. Without loss of generality, suppose $L(u, v_1) \leq L(u, v_2)$, now we only need to prove the inequality $w^{i+1,4}(u, v_1) \leq w^{i+1,4}(u, v_2)$ holds. Since $L(u, v_1) < L(u, v_2)$, there are two possibilities:

1. In the case $S_1 = S_2 = 4$ and $Y_1 = 0 < Y_2 = 1$, since these two items are in the same stage and have been exposed, we know $r(u, v_1) < r(u, v_2)$ and $p^{i+1,4}(u, v_1) = p^{i+1,4}(u, v_2)$, which are obtained after taking expectation over equations $Y_1 < Y_2$ and $O_1^{i+1,4} = O_2^{i+1,4}$. Therefore, we have $w^{i+1,4}(u, v_1) < w^{i+1,4}(u, v_2)$.

2. In the case $S_1 < S_2$. According to Assumption 2, we have $\frac{p^{i+1,4}(u, v_2)}{p^{i+1,4}(u, v_1)} \geq a \geq \max\{\frac{r(u,v_1)}{r(u,v_2)}, \frac{r(u,v_2)}{r(u,v_1)}\} \geq \frac{r(u,v_1)}{r(u,v_2)} \geq \frac{1}{a}$. Therefore, $p^{i+1,4}(u, v_1) r(u, v_1) \leq p^{i+1,4}(u, v_2) r(u, v_2)$, which is exactly $w^{i+1,4}(u, v_1) \leq w^{i+1,4}(u, v_2)$.

Thus we prove the first part of this theorem.

When $z_0 = z_1 = \cdots = z_4 = 0$ and $z_5 = 1$, this is the algorithm we mentioned at the beginning of Section 5, and the conclusion holds because $\mathbf{E}[L(u, v)] = \mathbf{E}[O^{i+1,4}(u, v) Y(u, v)] = w^{i+1,4}(u, v)$.

Now we finish the whole proof.

□

## A.4 Implementation Details of Prerank / Rank Model in Simulated Multi-Stage Pipeline

The settings of prerank / rank model in simulated multi-stage pipeline is listed in Table 7. The label in the multi-stage pipeline equals 1 when: play_duration >= video_duration if video_duration <= 7,000, or play_duration > 7,000 if video_duration > 7,000,

Table 7: Settings of prerank / rank in simulated pipeline.

| | prerank | rank |
|---|---|---|
| model | User DNN & Item DNN | DNN |
| hidden layers | [256, 128] | [1024, 512, 256] |
| optimizer | Adam | |
| learning rate | 0.001 | |
| batch size | 8192 | |

