# OpenReview forum: "Full Stage Learning to Rank: A Unified Framework for Multi-Stage Systems"
_ACM.org/TheWebConf/2024/Conference — TheWebConf24_

### Official Review · Reviewer_Fvwk · 2023-11-06

**Novelty:** 5
**Technical Quality:** 5

**Review:**

This paper focus on the joint optimization on multi-stage recommender systems (i.e., retrieval, preranking, ranking, reranking). The authors extend the probability ranking principle to generalized probability ranking principle (GPRP). A full stage learning to rank (FS-LTR) framework is designed, where the ranking function of non-final stage would consider both the user final feedback and the selection bias estimation for the subsequent stage. The experiments are provided on both one public dataset and industrial applications with AB test.

However, the joint optimization of multi-stage recoommender systems is not a brand-new story. There are a few existing works that focus on the same goal [1], but they are omitted in the paper (no discussion, no comparison).

[1] RankFlow: Joint Optimization of Multi-Stage Cascade Ranking Systems as Flows

**Questions:**

This paper is generally sound to me. I suggest authors provide further survey and disccusion on previous works that jointly optimize the multi-stage recommender systems.

**Reviewer Confidence:**

4: The reviewer is certain that the evaluation is correct and very familiar with the relevant literature

**Scope:**

4: The work is relevant to the Web and to the track, and is of broad interest to the community

---

### Official Review · Reviewer_m6nt · 2023-11-23

**Novelty:** 5
**Technical Quality:** 4

**Review:**

**Summary**

This paper challenges the widespread application of the Probability Ranking Principle (PRP) in multi-stage Information Retrieval (IR) systems due to inherent selection bias across stages. It introduces the Generalized Probability Ranking Principle (GPRP) and the Full Stage Learning to Rank framework, effectively addressing selection bias and enhancing performance in both retrieval and ranking stages. Extensive evaluations, including simulations and live A/B tests on a top short-video recommendation platform, confirm the algorithm's consistent and significant performance improvements post-deployment.

**Strengths**

- The problem of studying LTR in the context of full stage is important. This paper also proposes deep insights about the selection bias in the full stage LTR, which is natural and interesting.

- Methods are supported by theories.

- Experiments on simulation and online A/B tests show a strong improvement.

**Weaknesses**

This paper doesn't adequately cover the existing research on selection bias in LTR, such as

- Harrie Oosterhuis and Maarten de Rijke. 2020. Policy-Aware Unbiased Learning to Rank for Top-k Rankings. In Proceedings of the 43rd International ACM SIGIR Conference on Research and Development in Information Retrieval (SIGIR '20).
- Zohreh Ovaisi, Ragib Ahsan, Yifan Zhang, Kathryn Vasilaky, and Elena Zheleva. 2020. Correcting for Selection Bias in Learning-to-rank Systems. In Proceedings of The Web Conference 2020 (WWW '20).
- Zohreh Ovaisi, Kathryn Vasilaky, and Elena Zheleva. 2021. Propensity-Independent Bias Recovery in Offline Learning-to-Rank Systems. In Proceedings of the 44th International ACM SIGIR Conference on Research and Development in Information Retrieval (SIGIR '21).

and so on.

The connection between this paper and previous efforts lies in addressing selection bias in LTR. It's essential to ascertain if prior methods in handling selection bias can yield similar improvements in full-stage LTR or if they have limitations. I believe this submission should delve into a discussion and comparison of these prior approaches.

**Questions:**

- Table 2 references "Bold numbers". Where are they?

**Reviewer Confidence:**

3: The reviewer is confident but not certain that the evaluation is correct

**Scope:**

3: The work is somewhat relevant to the Web and to the track, and is of narrow interest to a sub-community

---

### Official Review · Reviewer_f2Cw · 2023-11-24

**Novelty:** 5
**Technical Quality:** 4

**Review:**

This paper presents Full Stage Learning to Rank, a unified framework for multi-stage information retrieval (IR) systems. It challenges the widespread use of the Probability Ranking Principle (PRP) across all stages and introduces the Generalized Probability Ranking Principle (GPRP) to consider stage-specific selection bias and user interests. The proposed algorithm effectively estimates selection bias and improves performance in both retrieval and ranking stages, as validated through online A/B tests in a short-video recommendation platform.

While the starting point and theoretical derivations in this paper are commendable, the experimental section falls short of expectations. Firstly, in the experiment setup (8.1.2), it states that the multi-stage pipeline includes retrieval, preranking, and ranking stages. However, the subsequent experimental section only covers retrieval and preranking, lacking experimental results related to the ranking stage. Secondly, the baselines in this paper are overly simplistic. As a work on joint optimization for mitigating selection bias,  it has not been compared with various efforts in correcting selection bias, nor has it been compared with multi-stage joint optimization approaches, such as RankFlow[1] and ICC[2]. Therefore, even if online A/B tests demonstrate some improvement, the absence of these baseline makes it challenging to convince readers that this method truly achieves what it claims.

[1] Jiarui Qin, Jiachen Zhu, Bo Chen, Zhirong Liu, Weiwen Liu, Ruiming Tang, Rui Zhang, Yong Yu, and Weinan Zhang. 2022. RankFlow: Joint Optimization of Multi-Stage Cascade Ranking Systems as Flows. In Proceedings of the 45th International ACM SIGIR Conference on Research and Development in Information Retrieval (SIGIR '22)
[2] Luke Gallagher, Ruey-Cheng Chen, Roi Blanco, and J. Shane Culpepper. 2019. Joint Optimization of Cascade Ranking Models. In Proceedings of the Twelfth ACM International Conference on Web Search and Data Mining (WSDM '19)

**Questions:**

Why was the experiment designed with retrieval, preranking, and ranking stages, yet only results from the first two stages were presented?

Why were baselines for correcting selection bias or multi-stage joint optimization, not compared?

**Ethics Review Description:**

/

**Reviewer Confidence:**

4: The reviewer is certain that the evaluation is correct and very familiar with the relevant literature

**Scope:**

4: The work is relevant to the Web and to the track, and is of broad interest to the community

---

### Official Review · Reviewer_BXca · 2023-11-27

**Novelty:** 6
**Technical Quality:** 6

**Review:**

In this paper, the authors devise a method to improve ranking performance in a multi-stage retrieval system by training each retrieval stage on the output of its later stages, via introduced labels. The paper investigates the approach theoretically, in offline experiments, and via AB-tests in a live system. In each instance, the authors show a significant positive impact of the suggested method and explore how to select the negative sampling during training optimally.

The problem of optimizing a multi-stage system is of large relevance as such a design is very common and the authors provide a well-rounded context for the problem and why it is relevant to consider on top of other biases such as the positional bias. In addition, the authors provide a thorough theoretical analysis to firmly establish the assumptions being made in the approximations and what the implications of these are. The offline evaluations convincingly show how the method yields improvements in practice and it is nice to see that these benefits also carry out in an online setting.

**Questions:**

In Table 5: the best result was found to be with 1,1,2 followed closely by 1,2,1. Why was 1,2,2 concluded here as it looks like this combination might perform even better? 1, 2, 3 was included, so this seems like an omission. Would it be possible to easily extend up to all sampling of up to 3?
In the online evaluation, it is stated that the results are significant. However, the level of noise in the system is not shown. Would it be possible to add data that backs up whether these results are statistically significant, e.g. via statistical measures?

**Reviewer Confidence:**

4: The reviewer is certain that the evaluation is correct and very familiar with the relevant literature

**Scope:**

4: The work is relevant to the Web and to the track, and is of broad interest to the community

---

### Official Review · Reviewer_7WeX · 2023-11-28

**Novelty:** 6
**Technical Quality:** 6

**Review:**

Pros:
1. Tackles an important problem in multi-stage recommendation systems regarding selection bias and the applicability of traditional ranking principles.
2. Proposes a novel Generalized Probability Ranking Principle (GPRP) that accounts for both user interests and selection biases.
3. Develops an efficient algorithmic framework Full Stage Learning to Rank (FS-LTR) to operationalize GPRP.
4. Provides proofs for key theoretical results.
5. Conducts extensive experiments on both offline simulation and online A/B testing that demonstrate clear performance gains.

Cons:
1. Assumption 1 about pointwise exposure probabilities seems quite strong, authors provide some discussion but more analysis may be needed.
2. While Assumption 2 seems reasonable, additional real-world verification beyond posterior CTR data could strengthen the claim.
3. Although assumptions' validity is tested to an extent, further studies analyzing how robust the method is when assumptions are relaxed would strengthen the empirical analysis.
4. Additional ablation studies as well as testing on more public datasets could provide more insights into FS-LTR's properties and broad applicability.

Overall, I found this to be a clearly written, technically strong paper tackling an underexplored but important issue in an increasingly relevant area. The proposed GPRP principle is insightful and the FS-LTR framework seems effective. With some additional discussion around the assumptions and potential limitations, I believe this would make a nice contribution.

**Questions:**

When deployed, are there concerns about FS-LTR introducing too much randomness that could degrade user experience?

**Reviewer Confidence:**

3: The reviewer is confident but not certain that the evaluation is correct

**Scope:**

4: The work is relevant to the Web and to the track, and is of broad interest to the community

---

### Decision · Program_Chairs · 2024-01-22

**Decision:**

Accept

**Comment:**

This work introduces a comprehensive framework for multi-stage information retrieval systems. The reviewers generally concur on the novelty and good quality of this work. The authors have effectively addressed the reviewers' inquiries in their responses.